# Common Causes for Veterinary Visits among Australian Wildlife

**DOI:** 10.3390/ani14182662

**Published:** 2024-09-13

**Authors:** Agnes Gårdebäck, Maja Joäng, Maria Andersson

**Affiliations:** Department of Applied Animal Science and Welfare, Swedish University of Agricultural Sciences, P.O. Box 234, S-53223 Skara, Sweden; agnes.gardeback@gmail.com (A.G.); majajoang@gmail.com (M.J.)

**Keywords:** wildlife, veterinary care, veterinary hospital, Australia

## Abstract

**Simple Summary:**

Australian wildlife is often harmed by human activities. Veterinary hospitals play a crucial role in caring for individual animals and contribute to the protection of vulnerable species and increased animal welfare. This study investigated the wildlife patients arriving at two Australian hospitals to better understand the typical wildlife patients that veterinary hospitals can expect to encounter. Birds are the most common wildlife patients, often admitted due to injuries from car accidents. The study found a high mortality rate after admission, especially for animals with trauma or disease, compared to orphaned animals. It can be concluded that there are a number of challenges in the rehabilitation of wildlife. These findings add to our understanding of the challenges of caring for and preserving Australia’s unique wildlife.

**Abstract:**

Human activities in Australia frequently harm wildlife in their natural environments. Veterinary hospitals play an important role in treating individual animals and safeguarding threatened species. The primary objective of this study was to investigate the demographic and clinical characteristics of wildlife patients admitted to veterinary hospitals in Australia. Data from two wildlife hospitals situated in the southeast region of Australia was used to analyse the characteristics of wildlife patients. Avian species constitute the predominant category of wildlife patients admitted to these hospitals (54% and 60%, respectively). However, a large seasonal variation was observed for all types of animals. Traumatic injuries represent the foremost cause for admission for all types of animals; however, reptiles (62%) and birds (56%) were overrepresented in the category. Car collisions emerging as the most frequently encountered source of trauma. Moreover, the study reveals a notable mortality rate in admitted patients, approximately 50%, with an unfavourable prognosis for patients admitted due to trauma or disease. In conclusion, wildlife rehabilitation clearly presents a number of challenges. We recommend limiting rehabilitation patients, especially orphans and those not needing veterinary care, to focus resources on animals in real need. This could improve care quality, conserve resources, and enhance survival and release rates.

## 1. Introduction

The provision of veterinary care for wildlife poses a unique area of challenges, different from the typical clinical practice carried out in domestic animal hospitals. These challenges stem not only from inherent species differences but also from the limited knowledge in relation to wildlife health care, along with the possible demanding requirements associated with rehabilitation. At the same time, many species have already gone extinct, and the rate of extinction is increasing for birds and mammals in Australia [1]. Wildlife injuries are often caused by humans, both directly and indirectly from human causes [2,3,4,5,6]. Today there are many non-profit organisations working to protect and help Australian wildlife [2,7], as well as veterinary staff at different hospitals. The animals that arrive at hospitals are usually brought in by wildlife carers or caring citizens [8].

Wildlife welfare is regulated on a national level through the Environment Protection and Biodiversity Conservation Act and on a regional level in each state and territory [9]. Most native animals are considered protected wildlife but there are a few exceptions. Dingoes, eastern grey kangaroos, common brushtail possums, Bennett’s wallabies, and Tasmanian pademelons are not always considered protected wildlife [10]. A licence or permit is required for the rehabilitation of wildlife in all states and territories except Western Australia [10].

Admitting wildlife to veterinary hospitals requires the ability to predict the various types of patients that may arrive and the expertise to meet their specific needs. A 2018 nationwide survey [11] found that the main challenges faced by veterinary hospitals in caring for wildlife were a lack of necessary knowledge and skills, along with limitations in time and resources. Similarly, in 2021 a survey [8] revealed that knowledge and skill gaps were the second most significant challenge, after financial constraints.

As of now, there are a few studies that describe the overall distribution of species and reasons for admissions among wildlife at veterinary hospitals in Australia. Existing research in this context has been relatively limited. For instance, a study from 2019 [5] primarily investigated the admission and outcomes of wildlife patients at the Australia Zoo Wildlife Hospital. Additionally, two other studies [2,7] focused on animals taken into the care of volunteer wildlife rehabilitation organizations. It is worth noting that animals under the care of wildlife rehabilitators may, but not necessarily, be subsequently referred for examination and treatment at a veterinary hospital. Thus, gathering admission data directly from veterinary hospitals is important for gaining a comprehensive understanding of the most commonly encountered wildlife patients and their associated medical needs.

Studies have examined the reasons for wildlife admissions to veterinary hospitals in Australia, focusing on specific species. For instance, there is research on koalas [12,13,14,15], monotremes [16], reptiles [4], black cockatoos [17], coastal raptors [6], and flying foxes [3]. However, many other species remain understudied.

Numerous studies highlight the substantial impact of increasing human populations and urbanization on Australian wildlife. In southern Queensland, anthropogenic factors, such as car collisions, entanglement, and domestic animal attacks, accounted for 80% of admissions to a wildlife veterinary hospital, with these cases exhibiting higher mortality rates than other causes [5]. The majority of injuries sustained by wildlife patients in a study by Tribe and Brown [2], whether directly or indirectly, were linked to human activities. This was also true for various species in other studies, including 59% of Australian coastal raptors [6] and 63.7% of grey-headed flying foxes [3].

Trauma emerged as the predominant reason for admitting wildlife to veterinary care. A national survey indicated that 82% of admitted wildlife patients suffered from traumatic injuries [11]. Other studies corroborated this trend, with trauma cited as the primary cause for admission in various species, such as echidnas (90.1%), platypuses (73.7%), reptiles (73%), black cockatoos (76.7%), and koalas (41% to 38%) [4,12,13,16,17].

The specific causes of trauma varied among species. For koalas, echidnas, and reptiles, car collisions, and domestic animal attacks were common contributors [4,12,13,16,18,19]. Platypuses often suffered entanglement in fishing lines [16], while flying foxes were frequently entangled in fruit netting or barbed wire [20]. For cockatoos, car collisions and gunshots were prevalent [17], while coastal raptors faced entanglement, bird attacks, or car collisions [6].

Koalas, commonly admitted for either trauma or disease, exhibited variations in prevalence depending on the study. Some studies emphasized trauma [12,14,18] while others highlighted disease [13,15]. Chlamydia infection emerged as a significant contributor to disease in koalas, with some trauma cases also displaying signs of the infection [14]. Chlamydia infection rates among mainland Australian koalas varied from 21% to 88% depending on the region [21].

Overall, when considering all wildlife patients, vehicular accidents and orphaned animals were common [2,5,7,11]. In one study, the most common reasons for admission were car collisions (34.7%), followed by orphaned animals (24.6%), signs of disease (9.7%), dog attacks (9.2%), entanglement (7.2%), and cat attacks (5.3%). Car collision victims and animals attacked by dogs or cats had the highest mortality rates, while orphans had the lowest [5].

Birds and mammals were the most common wildlife patients, with birds typically outnumbering mammals, according to a national survey [11]. However, another study found variations, with birds being more prevalent in New South Wales and Queensland, while mammals predominated in Victoria [2]. Similarly, studies in New South Wales indicated that wildlife admitted to veterinary care consisted of 56.2% birds, 29.4% mammals, and 13% reptiles [8] and that wildlife carers encountered 53.4% birds, 34.1% mammals, and 12.5% reptiles [7]. In southern Queensland, mammals were more commonly admitted into veterinary care, accounting for 51.1% of patients, followed by birds (35.2%), reptiles (14.4%), and amphibians (0.3%) [5].

Seasonal variations affected admission patterns, with reptiles more commonly admitted on dry, warm days when they are typically active [19,22]. Echidnas, platypuses, grey-headed flying foxes, and reptiles were more frequently admitted during summer [3,5,16,22], while koalas, birds, and mammals, in general, were more common in spring [5,6,12,13]. The total number of animals in need of care across all species also peaked in spring [5,7]. Orphaned animals were six times more likely to be taken into care during spring than winter, while vehicle accidents were more common in autumn and winter [7].

Some studies investigated the outcomes of animals admitted to care. Taylor-Brown et al. [5] found that the overall mortality rate among admitted patients was 57.4%, with amphibians experiencing the highest mortality and mammals the lowest. However, another study reported mammals to have the highest mortality rate [7]. For specific animal types, mortality rates were 57% for black cockatoos, 44% for raptors, 57.9% for platypuses, 50.5% for echidnas, 53.9% for grey-headed flying foxes, 65.5% for koalas, and 47.5% for reptiles [3,4,6,13,16,17]. Most animals that died at veterinary hospitals were euthanized, whereas some studies indicated that unassisted deaths were more common in volunteer wildlife care organizations [2,20]. Studies in veterinary hospitals found that 10% of black cockatoos, 9.2% of grey-headed flying foxes, and 7.4% of reptiles died without euthanasia [3,17].

Cooper and Cooper [23] emphasize the importance of considering the welfare of wildlife patients throughout the rehabilitation process. A cost-benefit analysis should be conducted for each case, and decisions regarding euthanasia should be made promptly in cases of poor prognosis [2]. Adequate resources, including facilities and personnel for acute veterinary care and rehabilitation, must be available. The risk of introducing new diseases into the wild fauna and the potential for transmission between wildlife patients and domestic animals in care must also be considered [2,24]. Wildlife workers should be cautious of zoonotic infections, such as Australian bat lyssavirus in bats and Q fever in kangaroos. Vaccination and proper hygiene practices are recommended [25,26].

Stress can have prolonged effects on behaviour and mortality rates [27,28]. Anaesthesia should thus be used for large or aggressive animals and during stressful or painful procedures [29,30]. Pain management is crucial, and the presence or severity of pain in wildlife is often underestimated [31,32]. Changes in behaviour should be evaluated during pain assessment, and knowledge of normal behaviours for different species is, therefore, important [29]. In summary, treating wildlife patients requires a comprehensive understanding of their unique needs, welfare considerations, and potential risks to both wildlife and humans involved in their care.

The primary objective of this study was to investigate the wildlife species commonly encountered at veterinary hospitals in southeast Australia. The study aims to give a better understanding of the distribution between different types of wildlife patients, instead of focusing on a singular species, in order to predict the most common patients and their needs. The aim was also to investigate whether the numbers and species of admitted wildlife patients to Australian animal hospitals have changed, or are continuously increasing. The following research questions were addressed:What types of wildlife are admitted to veterinary hospitals in southeast Australia?Is there a seasonal variation in admissions of different types of animals?What are the common reasons for admission of wildlife?Does the reason for admission differ between different types of animals?What is the outcome after veterinary treatment?

## 2. Materials and Methods

Two wildlife hospitals in Australia have provided data for this study, Byron Bay Wildlife Hospital in New South Wales and Adelaide Koala and Wildlife Centre in South Australia. They are both situated near large cities; Brisbane and Adelaide, and consequently situated in an area where the wildlife is clearly affected by human population. The data provided was a summary of each hospital’s medical records, prepared by each hospital as part of their record keeping. The data included the species of each patient, the reason for admission to the hospital, the outcome after care as well as a few other parameters that have not been included in this study.

Monthly data from Byron Bay Wildlife Hospital was available starting from April 2021 until December 2022, resulting in data from 21 months. Whereas monthly data from Adelaide Koala and Wildlife Centre was available starting from January 2018 until November 2022, resulting in data from 59 months. Data for each individual patient was available for Byron Bay Wildlife Hospital, for Adelaide Koala and Wildlife Centre monthly summaries for outcome, species, and reasons for admission were used.

The reasons for admission at the two hospitals were not comparable at the start of this study and were thus combined into fewer and wider categories. The data were sorted using Microsoft Excel 2016 (16.0.5448.1000), listing every unique admission reason used by each hospital and removing obvious misspellings. This resulted in 96 different categories used by Byron Bay Wildlife Hospital and 10 different categories used by Adelaide Koala and Wildlife Centre. These categories were compressed into wider categories in different steps, combining synonymous categories, and similar reasons for admissions, resulting in 6 matching categories for the two hospitals.

The categories used to describe reasons for admissions were “disease”, “toxin”, ”trauma”, “healthy”, “orphan” and “other”. The category “other” was used for descriptions that did not correspond to any of the other categories or that could fit into several categories. The categories “healthy” were only used by Byron Bay Wildlife Hospital and were used when an animal mistakenly was taken into hospital, however was considered healthy. “Orphaned” was only used by Adelaide Koala and Wildlife Centre. This might not accurately reflect the actual situation, the categories were included since they contributed to a large part of admitted animals at the associated hospital.

All patients could be divided into one of the following classes: amphibians (*Amphibia*), reptiles (*Reptilia*), birds (*Aves*), and mammals (*Mammalia*). When counting individuals belonging to a certain species, different common Australian names referring to the same species were combined into the same category. In some instances, when there was only one subspecies with habitats close to the relevant hospital, a wider category such as, for example, carpet python was combined into a more specific category like coastal carpet python. In all such cases, the merge was first confirmed as applicable by the relevant hospital.

To present the outcome for patients after admission, four similar categories were used by both hospitals: “Euthanized”, “Died”, “Put into care” and “Released”. Byron Bay Wildlife Hospital had additional outcomes listed during some months. These outcomes were placed in one of the original four categories, “Dead on arrival” was included in “Died” while “Released by care giver” and “Creche” were included in “Put into care” to represent the outcome of the hospital stay. Individuals where no outcome was described were excluded from the data.

The results are depicted in various graphs, accompanied by descriptive statistics. A two-sample *t*-test was employed to analyse the number of admitted patients across different animal categories between hospitals. Additionally, the seasonal variation of patient admissions was assessed using Spearman’s rank correlation.

## 3. Results

In total 2613 patients were admitted to Byron Bay Wildlife Hospital. The data were complete except for the outcome analysis where 10 patients were excluded due to missing data. The total number of patients admitted to Adelaide Koala and Wildlife Centre was 7091, whereas 7039 patients had a described outcome, 7091 patients had a described reason for admission and 7066 patients could be categorised as either amphibian, bird, mammal, or reptile.

Birds were the most common type of patient at both Adelaide Koala and Wildlife Centre (54%) and Byron Bay Wildlife Hospital (60%). While mammals were the second most common type of patient for both hospitals, the proportion of mammals was twice as big for Adelaide Koala and Wildlife Centre (41%) compared to Byron Bay Wildlife Hospital (19%). However, there was no statistical difference in the type of admitted patients between hospitals (*t* = 1.04, *p* > 0.05).The most common species for each animal type and hospital can be found in Appendix A.

There was clearly a variation over the seasons in the number of different admitted patients in both hospitals, Adelaide Koala and Wildlife Hospital (Figure 1) and Byron Bay Wildlife Hospital (Figure 2). However, there was only a statistically significant correlation between hospitals in relation to the seasonal variation for birds, not for other types of animals (Spearman correlation: 0.867, *p* < 0.05). The most common reason for admission was trauma for both hospitals as seen in Figure 3. The category “healthy” was only used by Adelaide Koala and Wildlife Centre and the category “orphan” was only used by Byron Bay Wildlife Hospital. These categories were included since they contributed to a large part of admitted animals at the associated hospital.

The category “other” was used for reasons for admission that could not be categorised under any of the previous categories. For Byron Bay Wildlife Hospital the dominant reason for admission was “misadventure”, which constituted 30% of the animals placed in this category, followed by “unknown”, which described 16% of the animals. For Adelaide Koala and Wildlife Centre, 65% of the animals in the “other” category had a reason for admission that was described as “other/combo” by the hospital, and 22% of the animals were described with “unknown” reason for admission.

Reptiles, shortly followed by birds, were overrepresented in the category “trauma” for both hospitals as seen in Figure 4 and Figure 5. Amphibians were however often admitted due to other reasons or because of disease. Adelaide Koala and Wildlife Centre had only one admission of amphibians in total and is therefore not included.

Car strikes were the most common reason for trauma among birds, mammals, and reptiles admitted to Byron Bay Wildlife Hospital. The trauma in amphibians was, however, often unspecified, as shown in Figure 6. Similar data on different causes of trauma was not available from the Adelaide Koala and Wildlife Centre.

The total mortality for patients included the euthanized animals, as well as the ones that died during hospitalising. The total mortality among admitted patients at Adelaide Koala and Wildlife Centre was 50%, close to that of Byron Bay Wildlife Hospital at 52%. At Adelaide Koala and Wildlife Centre, 12% of admitted patients were released, and 38% were put in care. At Byron Bay Wildlife Hospital 4% were released, and 44% were put in care.

The most common outcome differed between species admitted to Adelaide Koala and Wildlife Centre, as shown in Figure 7. Reptiles were much more likely to be released than other animals, mammals were commonly put in care and birds were often euthanized. The outcomes from Byron Bay Wildlife Hospital could not be identified in relation to every specific individual.

## 4. Discussion

This study showed that birds were the most frequently treated wildlife in the regions under investigation. This finding aligns with national data reported by Orr and Tribe [11] and is consistent with studies conducted in New South Wales by Tribe and Brown [2] and Haering et al. [8]. However, it is worth noting that research conducted in Victoria and southern Queensland has indicated that mammals are the most common wildlife patients in those areas, suggesting geographical variations in different populations [2,5].

There is clearly a large variation in the admitted patients, but also in the outcomes. The variance in admitted mammals between the two hospitals might possibly be attributed to these geographical differences. However, other factors like proximity to larger cities, the hospital’s specialization, the expertise of collaborating caregivers, and competitive situations can also influence the types of wildlife patients that are admitted for care. Koalas are for example much more likely to be admitted to Adelaide Koala and Wildlife Centre due to the name of the hospital.

The data revealed monthly fluctuations in admission numbers at both hospitals, with the most pronounced fluctuations observed in bird admissions at the Adelaide Koala and Wildlife Centre. These fluctuations exhibited peaks, particularly around November each year, which could potentially be attributed to fledglings leaving their nests during this period. A similar pattern in bird admissions was tentatively noted in the data from Byron Bay Wildlife Hospital. To gain a comprehensive understanding of these seasonal variations, more in-depth analyses are required.

It is important to note that the reasons for admissions do not necessarily correspond to the most common injuries in wildlife overall. There are several factors influencing whether the animal is admitted into care or not. Animals in a visible place close to humans are more easily found and some animals may be more popular than others in the eyes of the public and thus have a higher chance of receiving help. Injuries that cause the animal to die at the scene are also rarely admitted, such as amphibians hit by cars. In some cases, as illustrated in Figure 3, healthy animals are mistaken for being in need of care and brought to an animal hospital. This indicates that members of the public might find it hard to differentiate healthy animals from those in need of care, according to Figure 5, this problem is relevant to all different types of animals.

Trauma was the most common reason for admission to both hospitals. This is supported by previous research by Orr and Tribe [11] and for single species by Scheelings [16], Scheelings [4], Le Souëf et al. [17], Griffith et al. [12] and Burton and Tribe [13]. Trauma was the main cause for the admission of all species except amphibians. One explanation can be that there were few amphibians admitted overall compared to the other groups, making the results less reliable for amphibians. Another explanation can be that amphibians are less likely to be brought to a wildlife hospital after traumatic events, possibly because of the high mortality when affected by trauma. It is also possible that signs of trauma are not recognised in amphibians to the same extent as in other animals.

For birds, mammals, and reptiles, the most common reason for trauma was car strikes. This is supported by Taylor-Brown et al. [5], Orr and Tribe [11], Tribe and Brown [2], and Kwok et al. [7] as well as several studies on individual species. Another common cause of trauma in mammals was entanglement, possibly because of the large proportion of flying foxes in this group (see Table A3). Scheelings and Frith [3] and Mo et al. [20] have shown that entanglement is the major cause of trauma in flying foxes. Reptiles were instead overrepresented in the category “animal attack”, similar to results shown by Shine and Koenig [22], Koenig et al. [19], and Scheelings [4]. The fact that many reptiles are relatively small and ground-dwelling might make them more vulnerable to attacks compared to the other groups. Birds being the only animals commonly affected by window strikes was expected.

In the context of this study, it is essential to acknowledge that the categories “healthy” and “orphaned” were only attributed to one hospital, resulting in the other hospital’s admissions for these categories being recorded as zero, even if this might not accurately reflect the actual situation. One plausible theory is that at Adelaide Koala and Wildlife Centre, orphaned animals might have been categorized as healthy if there were no overt signs of admission-related issues or placed within the “other” category. Similarly, at Byron Bay Wildlife Hospital, healthy animals admitted could have been classified under “misadventure”, consequently falling into the “other” category. Therefore, it is important to interpret the results of this study with caution and not conclude that no orphans were admitted to Adelaide Koala and Wildlife Centre or that no healthy animals were admitted to Byron Bay Wildlife Hospital. The prevalence of orphaned animals as a reason for admission is supported by prior research conducted by Orr and Tribe [11], Tribe and Brown [2], Taylor-Brown et al. [5], and Kwok et al. [7]. In this study, the most frequently admitted orphaned animals belonged to the bird and mammal categories, which aligns with the findings presented by Taylor-Brown et al. [5].

Reptiles in the category “disease” were much lower for animals admitted to Adelaide Koala and Wildlife Centre compared to Byron Bay Wildlife Hospital. Again, this could potentially be explained by how the hospitals categorises admissions. Byron Bay Wildlife Hospital had a lot more subcategories falling under disease compared to Adelaide Koala and Wildlife Centre, creating a possible discrepancy between the hospitals. At Adelaide Koala and Wildlife Centre the category “combo/other” is expected to be common for reptiles with the disease since the disease itself might not be recognized by people. However, the disease can cause reptiles to be slower, become an easier target for animals, or end up on roads and thus be admitted to the hospital for other reasons than the underlying disease. When interpreting the results of this study reptiles cannot be assumed to be less affected by disease in one place compared to the other.

The outcome after admission was found to be surprisingly equal between the two hospitals and aligns with the results presented by previous research. Studies have found that the mortality of admitted wildlife spans from 45% to 65% and that between 7% and 10% of animals at veterinary hospitals die without euthanasia [3,4,5,6,13,16,17]. The mortality of around 50% shows that veterinary care for wildlife often consists of making the hard but important decision to euthanize wildlife without a good prognosis to thrive in the wild. On the other hand, there is a challenge to care for these animals, and most probably we are not providing adequate care for the wildlife patients. The high admission rate of trauma patients might further contribute to the mortality since trauma was found to often have a negative outcome. This, as well as the better prognosis for orphans, is supported by Taylor-Brown et al. [5].

Contrary to the situation at veterinary hospitals, two studies show that deaths at volunteer wildlife care organisations are often unassisted, i.e., without the use of euthanasia [2,20]. In the study by Mo et al. [20], the high number of deaths without euthanasia was largely influenced by bats admitted because of heat stress, 92 out of 1017 bats that died in this group were euthanized. The discrepancy, when compared to veterinary hospitals, is not necessarily caused by different assessments of the prognosis for the patient, it can also be caused by the fact that wildlife care organizations often see patients earlier than veterinary hospitals when picking up patients from the location of the incident. At hospitals, a decision to euthanize can be completed quickly, wildlife carers will however often need to travel to a veterinarian and might not have access to a veterinary hospital that is open around the clock. There are currently not enough studies to draw any conclusion as to whether the discrepancy exists or if it is a bias caused by differences in journal documentation. More research is needed in this area, understanding the underlying causes of unassisted deaths is important to mitigate unnecessary or prolonged suffering of animals in care.

It is important to note that the scope of this study was limited to the data obtained from two veterinary hospitals situated in the south eastern region of Australia. We do not know the reason behind the choice to go to one of the hospitals. There might be other factors like the name or the specialisation of the hospital. Therefore, further comprehensive research is required to gain a nationwide perspective on this matter and also in relation to the differences between hospitals in how they handle admitted patients and what type of care they are providing.

## 5. Conclusions

In conclusion, this study reaffirms findings consistent with prior research investigations. This study clearly highlights the challenges that animal hospitals face in the rehabilitation of wildlife. We would like to recommend limiting the number of rehabilitation patients, particularly orphans or those not requiring veterinary intervention, to focus resources on individuals who truly need care. This approach ensures better use of limited resources, improves patient care, and upholds ethical standards, rather than overusing resources and potentially compromising care, which can result in lower survival and release rates. The role of veterinary hospitals in the rehabilitation of wildlife needs to be addressed further and integrated into broader conservation efforts to ensure the effective treatment and release of injured animals back into their natural habitats.

## Figures and Tables

**Figure 1 animals-14-02662-f001:**
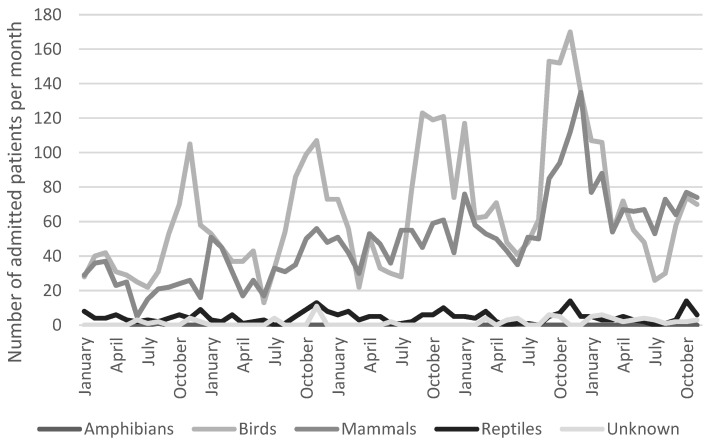
Seasonal variation among patients admitted to Adelaide Koala and Wildlife Centre (*n* = 7091).

**Figure 2 animals-14-02662-f002:**
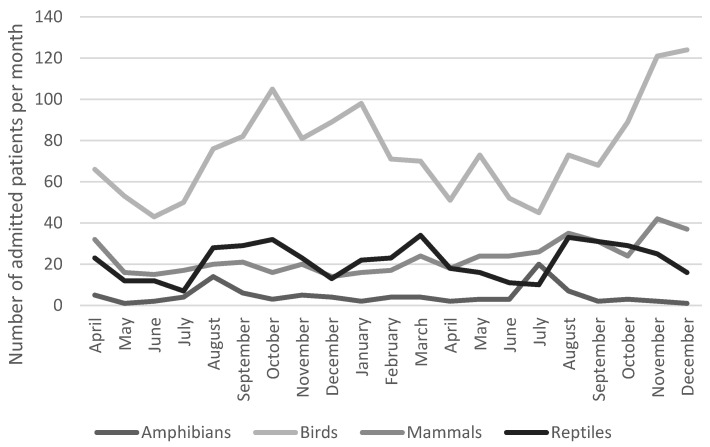
Seasonal variation among patients admitted to Byron Bay Wildlife Hospital (*n* = 2613).

**Figure 3 animals-14-02662-f003:**
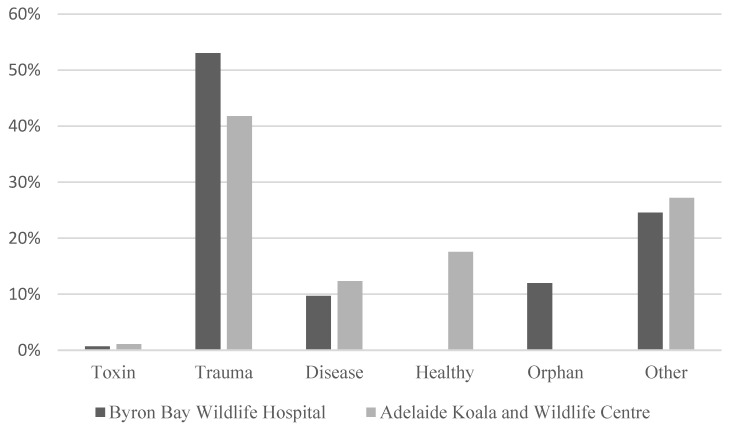
Reasons for admission of wildlife animals to the two clinics. A total of 2613 individuals were at Byron Bay Wildlife Hospital, and 7091 individuals were at Adelaide Koala and Wildlife Centre.

**Figure 4 animals-14-02662-f004:**
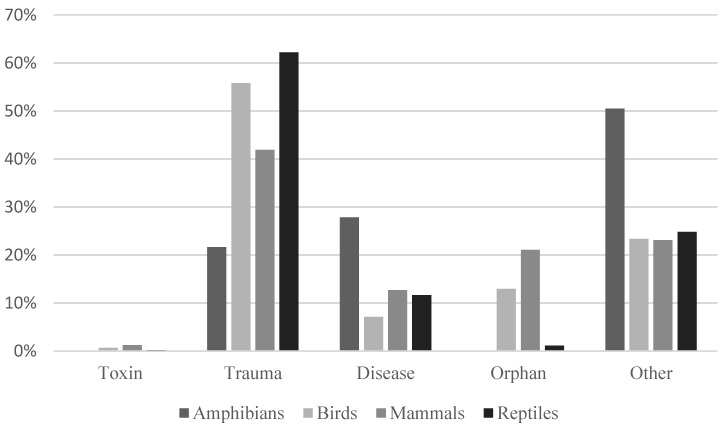
Reasons for admission of wildlife animals in relation to type of animal at the Byron Bay Wildlife Hospital (*n* = 2613).

**Figure 5 animals-14-02662-f005:**
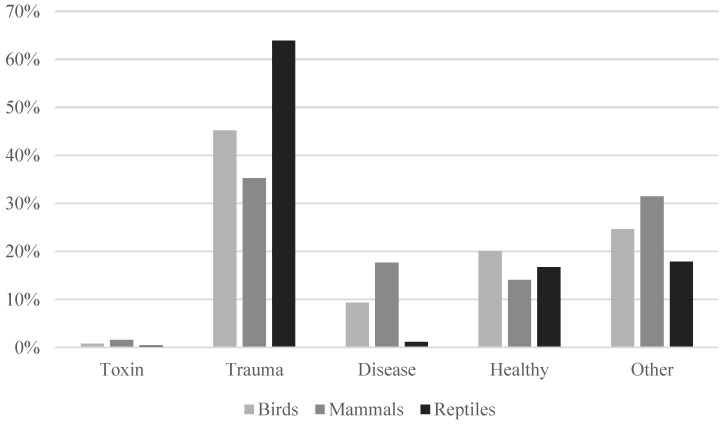
Reasons for admission of wildlife animals in relation to type of animal at the Adelaide Koala and Wildlife Centre (*n* = 7091).

**Figure 6 animals-14-02662-f006:**
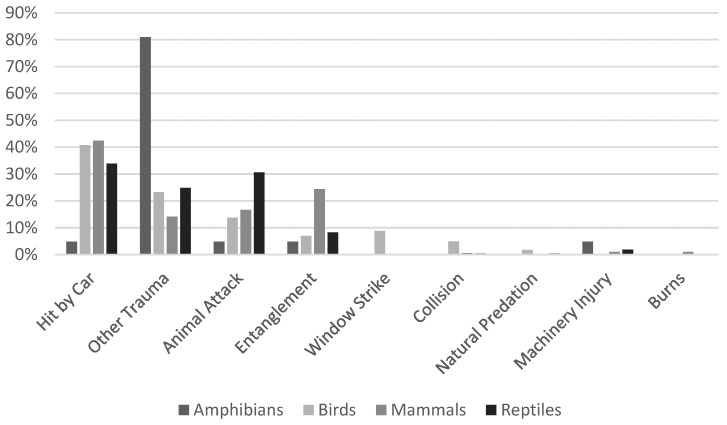
Causes of trauma in patients admitted to Byron Bay Wildlife Hospital (*n* = 640). Data in the category “other trauma” was not specified.

**Figure 7 animals-14-02662-f007:**
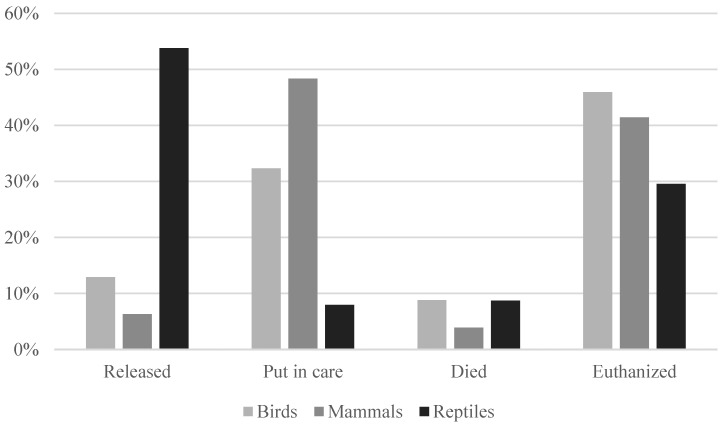
The outcome of admitted patients at Adelaide Koala and Wildlife Centre (*n* = 7039).

## Data Availability

The data presented in this study are available on request from the corresponding author, as the involved animal clinics want to be informed about the distribution.

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
