# Peer review of "Common Causes for Veterinary Visits among Australian Wildlife"

_animals, 2024, doi:10.3390/ani14182662_

Round 1

Reviewer 1 Report

Comments and Suggestions for Authors

In this study the authors collected data from two veterinarian hospitals from the southern region of Australia to investigate the characteristics of the wildlife admitted as patients and the outcome of these admissions.

My main comment is that there are already numerous similar studies and the authors list these in their very comprehensive introduction. Therefore, it is not really clear to me what is the main purpose of their study, or even if there is one at all, besides repeating what has already been described elsewhere. The results are not novel and quite expected given the type of data. The conclusions support previous studies as repeated several times by the authors, but this is simply because it seems that nothing new was added to this study. There is the need to clarify 1) why was the study conducted and 2) what is novel about it, is it adding something new to what we already know?

Line 23: delete (data)

Line 137: looks like the main cause of death is the inability to provide adequate care

Lines 167-174: Your introduction is very comprehensive and describes a number of studies that tackled the same research questions you are investigating. For this reason I do not fully understand the objective of this study and the rationale for asking similar (if not the same) research questions of previous studies. Please clarify

Line 187: add "whereas for Adelaide "

Line 194: please write the categories

Line 196-198: would make more sense to write "the patients admitted to the hospitals belonged to three classes"

Line 199; Do you mean common Australian names? Plover and masked lapwing are two very distinct species to me

Lines 212-219: It will be easier to read if you only listed the total number of patients for each hospital that made it into your final data set

 Line 223: I strongly doubt this is because of a real geographic differences in fauna composition, rather I suppose that the name and specialisation of the hospital matters. I assume that, if possible, people would more easily bring a mammal to a "Koala Center " than to a raptor center or any general veterinary clinic.

Lines 232- 234: I don't fully understand the use of these two categories. First, if the animal was healthy, why was it then at the hospital? Second, you could include orphans in any of the other categories depending on their health status when they arrived at the hospital : were they sick, healthy or had a trauma?

Line 239: It would be easier to merge the two graphs into one, similar to Figure 3. You would only need to collapse the category " Healthy" and " Orphan" into more appropriate ones

Figure 6: it would be clearer to use “other” rather than “trauma”

Figure 8: the legend should probably read “Byron Bay”. Figure 7 and 8 could also be merged into one figure

Lines 271-273: it is not clear to me why there should be such a variation. It would be useful to expand on this

Line 277: as I mentioned earlier, I suspect that hospital specialisation is actually the main reason for the difference. For example, I would expect than a hospital called “Koala Center” is quite specialised in mammals rather than birds…

Lines 283-285: to me is completely expected as most birds in Australia breed between August and December. The seasonal increase would thus include young chicks, fledglings but also adults that are more mobile during the breeding period

Lines 321-323: this is why you should have a better category for these animals in your data

Lines 343-343: do you mean “not euthanised”? Or that they were not provided with any care at all?

Line 359: delete “all”

Line 370: I am still not quite sure about the main purpose of this study as it seems to me that it just repeats results of previous research without adding any new info…

Author Response

Thank you for giving constructive comments on the manuscript. Here is our response. Best wishes

Reviewer 2 Report

Comments and Suggestions for Authors

Dear Authors,

The manuscript is describing an interesting and important topic. It is well-structured, easy to read and follow. However, I feel that this version still requires improvements in the following points:

- providing explanations why it was important to perform this analysis on these datasets

- improving the description of the methods for some details

- including statistical analyses comparing animal groups - seasons - causes / outcomes

- including species-level information in the Results and Discussion

- providing take home messages what to do with these results, how to use them in the practice

Please, check my comments in the uploaded pdf file.

Author Response

(The authors gave the same response as above.)

Round 2

Reviewer 2 Report

Comments and Suggestions for Authors

Author Response

Dear reviewer, 

thank you for helping us to further improve our paper, accordingly:

1) a number of spelling mistakes are now corrected.

2) The result section is completed with a statistical analysis between the type of admitted patients per hospital. There is also a correlation analysis between hospitals in relation to the number of admitted patients per month.

Thank you for again helping us to improve the paper.

Best wishes

Maria Andersson